# 3D Polymer Architectures for the Identification of Optimal Dimensions for Cellular Growth of 3D Cellular Models

**DOI:** 10.3390/polym14194168

**Published:** 2022-10-04

**Authors:** Christian Maibohm, Alberto Saldana-Lopez, Oscar F. Silvestre, Jana B. Nieder

**Affiliations:** INL—International Iberian Nanotechnology Laboratory, Ultrafast Bio- and Nanophotonics Group, Headquarters at Av. Mestre Jose Veiga, 4715-330 Braga, Portugal

**Keywords:** 3D cell scaffolds, two-photon polymerization, spectral imaging, linear-unmixing, time-lapse studies

## Abstract

Organ-on-chips and scaffolds for tissue engineering are vital assay tools for pre-clinical testing and prediction of human response to drugs and toxins, while providing an ethical sound replacement for animal testing. A success criterion for these models is the ability to have structural parameters for optimized performance. Here we show that two-photon polymerization fabrication can create 3D test platforms, where scaffold parameters can be directly analyzed by their effects on cell growth and movement. We design and fabricate a 3D grid structure, consisting of wall structures with niches of various dimensions for probing cell attachment and movement, while providing easy access for fluorescence imaging. The 3D structures are fabricated from bio-compatible polymer SZ2080 and subsequently seeded with A549 lung epithelia cells. The seeded structures are imaged with confocal microscopy, where spectral imaging with linear unmixing is used to separate auto-fluorescence scaffold contribution from the cell fluorescence. The volume of cellular material present in different sections of the structures is analyzed, to study the influence of structural parameters on cell distribution. Furthermore, time-lapse studies are performed to map the relation between scaffold parameters and cell movement. In the future, this kind of differentiated 3D growth platform, could be applied for optimized culture growth, cell differentiation, and advanced cell therapies.

## 1. Introduction

Animal models have long been the golden standard for the understanding of diseases, and their progression, as well as for physiology and drug development. The downside of using animal models is the often found discordances between animal and human studies, rendering results either only partly usable or downright wrong and non-transferable [1,2]. To replace animal models and create more representable models, small physiological systems often called organ-on-chips (OOCs) are designed and fabricated. OOCs can be seen as mimicking representations of full or specific parts of human organs, and are made by combining cell biology with advanced engineering and fabrication techniques. OOCs are set out to incorporate the 3R’s (reduce, refine and/or replace animal models), and through their functionality and design they provide insights into organ function, disease pathophysiology, drug development as well as more accurately predict the safety and efficacy of drugs [1,2,3]. To acquire the desired functionality, OOCs are commonly lined with the appropriate living cells for replication of the specific organ and function. The composition and 3D arrangement of the cells, usually provided through an extracellular matrix (ECM), are influential for survival, the needed polarity and morphology [4,5]. Reverse-engineering and replicating a full organ requires that the OCCs harbor multiple cell types in the right configuration, and are therefore complex to fabricate and seed with multiple cell lines. As so far no single OCC system has been able to completely mimic all functionalities of a full organ [1].

A simpler, but still very important type of OCCs, are single cell line OCCs, used mainly for toxicity assessments [6], where the single cell line in these models mimics the barrier interesting for the specific toxicity study. An example is the barrier found in the respiratory system of the human lungs, which provides the interface between an external environment and the systemic, i.e., gas transportation in the body, while also serving as a barrier to toxins and pathogens [7,8,9,10]. The first interface or barrier found in the lung system is an epithelia cell layer which will be the basis of most lung models, and most inhaled medications are therefore targeting cells in this layer. To design a model replicating this first barrier, a fundamental knowledge of not only the individual cellular behavior but also how this behavior is influenced by the 3D ECM is needed. The ECM is a complex structure consisting of proteins and other molecules surrounding the tissue cells, thereby providing the cell with both a supporting and shaping function.

Replicating such a function requires a fabrication method where, the 3D architecture of the ECM of the model can be precisely controlled and easily adapted for specific purposes. Such a model would incorporate morphological hierarchy in 3D, with feature sizes spanning from sub-cellular to large tissue and organ parts on the mm and cm scale.

Standard light and e-beam lithography techniques can structure surfaces covering resolution scales from a few nm to hundreds of µm in many different materials [11,12]. These methods are generally limited to 2 and 2.5 D structuring in a single process step, and design changes are not easily integrated and updated. Additive manufacturing is another fabrication pathway that has widely been used to structure sample surfaces in 2, 2.5 and 3D with different levels of complexity for bio-essays, tissue engineering, and organoids [13,14,15,16,17], and spans the wanted feature sizes. Additive manufacturing can be done by a variety of different fabrication methods including foam printing [18], bioprinting [19] and electrospinning [20]. Another sub-category of additive manufacturing is the non-linear optical process of two-photon polymerization (TPP), where in a single fabrication step free-form 3D structures with varying feature sizes can be fabricated in a variety of materials [21,22]. The adaptability of design, material, and relatively fast prototyping of the TPP process is ideally suited for the fabrication of structures for in vitro and pre-clinical cell interaction studies and tissue engineering.

Here we report on the effects of controlled and varied 3D cell niche structures on the distribution and movement of A549 epithelia cells. The structures are fabricated via TPP from a low shrinkage inorganic-organic bio-compatible photoresist SZ2080, and are meant to replicate the purely structural part of the ECM functionality. The structures also incorporate a sometimes overlooked property of scaffold systems, namely the ease of imaging, where 2D models are relatively accessible, and information extraction from 3D models becomes increasingly complicated. The presented scaffold structures are therefore designed as vertical walls facilitating easy imaging via spectral separation confocal fluorescence microscopy, where varying niche size along the wall structures incorporates the needed 3D morphological hierarchy. The variation in scaffold parameters leads to a differentiated distribution of the incubated A549-cells across the scaffold, which is analyzed on the basis of the parameter changes. The same 3D scaffold design is used in time-lapse studies where cellular motility is again related to scaffold morphology parameters. This kind of differentiated 3D growth platform, could find application in the optimization of culture growth, cell differentiation and advanced cell therapies.

## 2. Materials and Methods

### 2.1. Structure Design

The presented scaffold design, previously introduced in [23], encompasses two distinct main features, namely morphological hierarchy and keeping a fully vertical design for easy imaging. We base the design around a unit wall structure which is replicated in different configurations to create the full scaffold design. The unit wall structure is a 210 µm long and 24 µm tall (red), where the wall is divided into three sections of 10 µm, two sections of 30 µm, one section of 50 µm, and one section of 70 µm, as seen in Figure 1a. The sectioning is made by 15 µm cross walls (green), with 7.5 µm on each side of the long wall creating what we define as “niches” on both sides of the wall, where an example of 7.5 × 30 µm is indicated, as the blue area in Figure 1a. The unit wall structure is replicated 4 times, along the direction of the wall, where the last two unit walls (unit 3 and 4) are mirror images of the first two, see Figure 1b. This creates a stable 840 µm long wall structure, where the niche sizes gradually increase, changing from 10 to 70 µm, but also features large size jumps back from 70 to 10 µm. All structures are constructed from single traced lines, with a Z-slicing of 4 µm between each layer. The scaffold structure design for TPP fabrication was made directly in the µFab software version 3.9.8 as part of the Newport software, supporting the µFab microfabrication laser work station.

For further incorporation of morphological hierarchy in the scaffold structure, the 840 µm wall structures were fabricated next to each other with an inter wall distance, which we hereafter call wall separation. An example of a 20 µm wall separation, shown between 210 µm unit sections is seen in Figure 1c. The two parallel walls with niches create small chambers which are open on both ends, an example is indicated by the light blue square in Figure 1c. For the full scaffold structure, the 840 µm wall is replicated 16 times, in sets of 4 walls with 20, 25, 35, and 55 µm wall separations, respectively, as seen in Figure 1d, creating a scaffold with a volume of 840 × 520 × 24 =10.5 × 10^6^ µm^3^. In the designed scaffold the footprint of the smallest chamber size is 10 × 20 = 200 µm^2^, while the largest chamber size is 55 × 70 = 3850 µm^2^. All scaffold dimensions are defined from the center of a traced line as indicated at the top of Figure A1a, and are defined by the fabrication stage accuracy (1 nm in XY and 20 nm in Z). This will not introduce any measurable deviation in the sizes for the scaffold dimensions, however, the extension of the fabrication volume creates a thickness in XY of the lines which are measured by SEM to be approximately 2 +/− 0.2 µm, as seen in Figure A1a. This reduces all niche sizes by approximately 2 µm, and elongates all line endings by approximately 1 µm. In Z the height of the scaffold is measured to be 24 +/− 0.2 µm, as seen by the constant scaffold height in Figure A1a. Comparing this to the size of the A549 cells, which is estimated to be between 10 and 20 µm in diameter, as seen in Figure A1b, and assuming a circular cell shape would give an area between 314 to 1256 µm^2^. For an average cell size, we assume a value in the middle at 15 µm, in good accordance with previously reported values from SEM images [24]. This value fits perfectly into the size range of the chamber sizes in our scaffold design, and for comparison, two cells with a diameter of 15 µm, are represented by the blue spheres in Figure 1c. This value fits perfectly into the size range of the chamber sizes in our scaffold design. For comparison, two cells with a diameter of 15 µm, are represented by the blue spheres in Figure 1c. The different wall separations together with the cross walls create chamber openings, which vary in size from 5 µm, considered sub-cellular size, up to 40 µm which is larger than the average cell size. This will provide openings from where the cell needs to alter its morphology to enter the niche, up to an opening size considered non-influential on cell morphology. These openings should therefore have an influence on the general motility, along the walls of the cells in the scaffold volume. In the current scaffold design, the cellular movement is restricted along the 840 µm walls, and the cells are assumed not to cross the solid walls. In future designs, the wall of course can be made passable by including openings, as provided in mesh based designs [14,25].

### 2.2. Two-Photon Polymerization Setup, Sample Preparation and Characterization

Structure fabrication is performed in a fixed beam sample scanning laser microfabrication workstation (µFAB, Newport), where a schematic representation of the different components in the setup is seen in Figure 2. As an excitation source for the TPP process a Ti:Sapphire based femtosecond (fs) laser (Tsunami, Spectra Physics), with a repetition rate of 80 MHz and tuned to 795 nm was used. Pulse control and optimization was achieved by the use of an external prism compressor and power control via a half-wave plate combined with a Glan-Thompson polarizer. A fast shutter controls the on/off action of the laser and the fabrication process, and a beam expander is used to overfill the back focal aperture of the 40× Nikon dry objective microscope (NA 0.75, 40×) used for fabrication. The sample is moved by translation stages which provide 1 nm resolution in X-Y and 20 nm in Z. The fabrication process can be monitored in-situ via a CMOS camera with a yellow light LED for illumination. Device control, such as translation stages, power control, shutter, CMOS camera and LED lamp is controlled through an XPS motion controller (Newport) via the µFAB software (Newport).

All presented structures are fabricated on cleaned #1.5 glass coverslip microscope slides. The coverslips are heated, prior to fabrication for an hour at 100 °C on a hotplate to remove water and solvent residues. 20 µL of the low-shrinkage hybrid organic-inorganic zirconium containing sol-gel polymer SZ2080 (Maria Farsari, IESL-FORTH, Heraklion, Greece) is drop-casted on the glass slide and subsequently baked for another 30 min at 100 °C [26]. After baking the sample is transferred to the µFAB writing station and the scaffold design is traced in the polymer layer.

Based on previous experiments with the SZ2080 polymer, the fabrication speed was set to 25 µm/s with an average laser power of 14 mW [25]. After the design has been traced, the sample is put for 45 min into a 1:2 solution of 4-Methyl-2-pentanone and 2-Propanol for development, with subsequently rinsing in ethanol and air drying.

#### Optical and SEM Imaging

For wide-field optical inspection of the fabricated scaffold structures, an upright Nikon Eclipse LV100 ND was used with a 10× objective in transmission mode.

SEM imaging (FEI NovaNanoSEM QUANTA 650FEG) was used for 3D characterization of the structures. Before SEM imaging 10 nm of gold was sputtered on the samples (Kenosistec—UHV multitarget confocal sputter).

### 2.3. Reduction of Polymer Auto-Fluorescence

The strong inherent fluorescence of the SZ2080 polymer is quenched, before the cell experiments by UV light treatment. The treatment consists of extended illumination of the structures with the light from a mercury vapor short-arc lamp in wide-field illumination mode through a 10× objective (Zeiss DAPI filter set 49, Ex: G 365, BS: 395, Em: 445/50). The field of view of the 10× microscope objective ensures uniform UV treatment of the whole scaffold structure. To establish the reduction in auto-fluorescence as a function of illumination time, a sequence of confocal scans was made with a 15 min interval during the quenching process, using a Zeiss LSM 780 confocal microscope with a 405 nm excitation laser through the same 10x dry objective.

### 2.4. Cell Culture and Staining of A549 Cells

A549 human lung carcinoma cell line was grown under standard culture conditions on DMEM 4.5 g L^−1^ glucose medium supplemented with 1% penicillin/streptomycin (Corning Cellgro) and 10% Fetal Clone III Serum (GE Healthcare). Phosphate-Buffered Saline (PBS) without calcium and magnesium was used to wash the cells and 0.25% trypsin-EDTA solution (Corning Cellgro) to detach cells from the culture flask. The cells were seeded inside a silicone well insert (2.0 × 10^5^ cells mL^−1^) encompassing the TPP scaffolds on the glass coverslip under sterile conditions and incubated under 5% of CO_2_ at 37 °C. After 24 h, the cell nuclei were stained with Hoechst 33342 (diluted 1:2000) (Thermofisher) and the cytoplasm membrane stained with CF488 conjugated to Wheat Germ Agglutinin (WGA) (Biotium), both incubated for 30 min. The cells were washed two times with PBS and fixed with 4% paraformaldehyde. The time-lapse studies followed a similar procedure, except the cells were seeded at a lower density (5.0 × 10^4^ cells mL^−1^) and were stained only with Hoechst 33342, followed by PBS washing and the well replenished with complete DMEM medium without phenol red for live cell imaging.

### 2.5. Confocal Imaging

A Zeiss LSM 780 confocal microscope with 405 and 488 nm laser excitation lines was used for confocal fluorescence imaging. Each confocal fluorescence image Z-stack consists of 12 slices with a 2 µm step size, covering the full 24 µm height of the wall structures. After seeding with A549 cells, the samples were transferred to an incubator for 24 h. For imaging, the samples were transferred to the confocal microscope stage inside an incubator at 37 °C and 5% CO_2_. Either Z-stack scans of the sample were performed, or as in the case of the time-lapse study, confocal fluorescence images were taken at approximately an hour interval, with 6 total time points measured for a total time of 5 h and 25 min.

### 2.6. 3D Hyperspectral Image Analysis via Spectral Linear Unmixing

For hyperspectral imaging experiments, we used laser excitations at 405 and 488 nm and recorded fluorescence images in the so-called lambda mode. We record fluorescence signals either in a spectral range from 416 to 687 nm (at 405 nm excitation) or from 494 to 687 nm (at 488 nm excitation). Each detection channel covers a spectral range of 9 nm covering 32 and 23 detection channels for excitation at 405 and 488 nm, respectively. For spectral linear unmixing, an algorithm, as part of the Zeiss Zen software was used, where representative regions of interest for the scaffold, cells and background were manually chosen for the algorithm. This allows us to separate the scaffold auto-fluorescence contribution from the fluorescence of the H33342-stained A549 cell nuclei in the case of excitation at 405 nm and the CF488-stained cytoplasm cell part in the case of excitation at 488 nm.

In the time-lapse studies, the obtained confocal image Z-stacks were further analyzed in the ImageJ-Fiji software to determine individual cell nuclei positions inside the scaffold structures as a function of time. Based on the relative cell size compared to the wall height, we chose to divide the scaffold into two sections for the analysis, the bottom part of the wall and the top part of the wall. The bottom part consists of the first 7 slices, i.e., 14 µm in the Z-stack which coincide with the average A549 cell size, and the top position covers the last 5 slices, i.e., 10 µm. Each of the individual slices, in both the bottom and top part are aligned, which is easily done using the wall structures as markers, and then each of the two stacks are projected, via the concatenate function in ImageJ, into one image for each of the two positions. This procedure is done for each of the 6 time points, creating 12 images where distances in the images are set by the scale bar option in ImageJ. A custom Matlab algorithm, see Appendix A, is used to analyze cell movement between time points, creating a gradient map for the full time-lapse study. The gradient map is used to relate different scaffold regions to the overall cell motility whereby the impact of morphological hierarchy inside the scaffold volume can be studied.

## 3. Results and Discussion

### 3.1. Development of 3D Scaffolds

The full TPP-fabricated scaffold is shown in Figure 3a via a SEM image, showcasing the morphological hierarchy throughout the scaffold and fully replicating the 3D design shown in Figure 1d. Figure 3b shows the unit wall structure with the cross walls defining the niches, as described in Figure 1a, while Figure 3c,d shows a rotated view of the scaffold, highlighting the different chamber openings and sizes. From the 3D SEM characterization, we find that all scaffold parameters are a perfect match to the design parameters, and that the integral scaffold resembles the 3D design representations shown in Figure 1.

### 3.2. Reduction of SZ2080 Polymer’s Auto-Fluorescence

Polymers used for TPP, are usually a mixture of monomers, oligomers and photo-initiators (PIs), where the PIs promote crosslinking by radical polymerization [27]. The PIs, such as Irgacure 369 in the case of SZ2080, express strong auto-fluorescence even in the final developed TPP structures, which can hinder the use of such structures in fluorescence microscopy. Reduction of scaffold auto-fluorescence is therefore important, and it has been reported that chemical treatment of scaffold structures either before cell seeding [28] or directly mixed into the polymer before fabrication [27], can pose an efficient way to reduce autofluorescence while maintaining bio-compatibility after the treatment.

Here, we show an alternative, and a non-chemical way for the reduction of the scaffold’s auto-fluorescence to a level where parallel fluorescence microscopy of cells is feasible, and no changes to the bio-compatibly of the structures are observed. The strong auto-fluorescence of the SZ2080 polymer, see Figure 4a, is reduced down to only 22% of its starting intensity using non-invasive UV exposure, see Figure 4b. For the UV-light treatment, the 365 nm emission of a Mercury arc lamp, was selected spectrally by filters and focused through a 10× objective for wide-field illumination in a microscope setup. The sample was exposed to the UV-light treatment from the top continuously for several hours. The integrated fluorescence of the scaffold, in subsequent fluorescence image scans, shows that the average intensity is decreasing with the duration of the UV-treatment, and the trend follows an exponential decrease in the tested time interval, as observable in the log-scale representation of the data shown in Figure 4b.

While the fluorescence intensity of the scaffolds allows for easy non-invasive optical visualization of the 3D architecture of the fabricated structures, it may pose a challenge to perform simultaneous cellular bioimaging. If the scaffold fluorescence intensity is too strong and saturates the detector, especially at the region of interest, namely the cell –scaffold interface, vital information is lost. Via the presented UV-light treatment it is possible to tune the auto-fluorescence to the desired level for cell studies, where the lowered auto-fluorescence of the scaffold still can be used as a position marker to identify cell position with respect to the scaffold.

### 3.3. Analysis of Scaffold Influence on Cell Packing

Here we present a 3D hyperspectral image analysis to study how the designed scaffold structure influences the packing of the A549 cells. We can use the lowered remaining auto-fluorescence contribution of the scaffold as position markers. We are able to identify and remove contributions stemming from the scaffold from the acquired image stack taken on scaffolds incubated with the double-stained A549 cells with Hoechst 33342, labeling the nucleus and CF488 labeling the cytoplasm. The 3D confocal fluorescence image stack recorded under 405 nm excitation is shown in Figure 5a after applying an intensity thresholding that reveals that even after 150 min of UV-light treatment, the scaffold’s auto-fluorescence is considerable stronger than the fluorescence contribution from the stained A549 cells. To be able to observe the cells the scaffold areas had to be leveled to saturation. To separate the fluorescence contributions from scaffolds, with labeled nuclei and cytoplasm, respectively, hyperspectral image stacks were recorded at 405 nm excitation and at 488 nm. Next, three region of interests (ROIs) were selected that represent (i) the auto-fluorescent scaffold, (ii) the labeled cells (in case of 405 nm excitation revealing the cell nuclei) and in case of 488 nm excitation revealing the cytoplasm) and (iii) the background, respectively for both stacks. From the selected ROIs the spectral contributions in the 32 detection channels (nuclei, excitation 405 nm), see Figure 5b and the 23 channels (cytoplasm, excitation 488 nm), see Figure 5c, are determined. The auto-fluorescence of the scaffold in both figures, are still strong and show a broad spectral contribution throughout all 32 and 23 detection channels, respectively.

Via the selected ROIs and the spectral linear-unmixing applied to the two image stacks we are able to identify and separate the contribution from the scaffold (as extracted from the 405 nm excitation stack), see Figure 5d, the cell nuclei, see Figure 5e and the cytoplasm region see Figure 5f, in the confocal Z-stack. A representative single confocal image slice of each of the three contributions is shown in Figure 5g–i, where it is clearly observable that the long scaffold walls have been removed from the images. The representative slides show a scaffold area of 520 × 630 µm which is used in the analysis and corresponds to the field of view of the 10× microscope objective.

Further zooming into the extracted separated 3D image stacks allows a more detailed evaluation of the efficiency of spectral separation and impressive accuracy of the applied spectral linear unmixing can be appreciated, where successful separation into scaffold structure revealing the different niche sizes, the cell nuclei and cytoplasm regions are clearly visible (see Figure 5j–l).

With the extracted cellular contents information from the spectral unmixing, it is now possible to analyze the influence of scaffold morphology, i.e., wall separation and niche size on cell distribution across the scaffold. In this first analysis of a static condition, where the emphasis is on the separation of fluorescence signals from the different cell parts and the auto-fluorescence of the scaffold, no data is extracted regarding the proliferation of the cells inside the scaffold area.

First, we analyze the number of nuclei and cytoplasm material per unit area found in the scaffold, as defined by the two scaffold parameters. Secondly, we analyze how the nuclei and cytoplasm material are distributed inside the chambers, i.e., if the cells are either in contact with the wall or located in the free space between the walls again as a function of the same two scaffold parameters.

In Figure 6a,b, the total amount, defined by pixels intensity of nuclei and cytoplasm fluorescent material, respectively, is summed in each chamber over the full height of the scaffold, and normalized to the area of the smallest chamber size of the scaffold, i.e., 10 × 20 = 200 µm^2^. For both cellular components, a similar monotonously decreasing trend of material presence as a function of wall separation is observed for all niche sizes. For both cases, the nuclei and the cytoplasm, the curves converges towards a common value for the different types of cellular material, which is independent of the niche size.

In Figure 6a, when the wall separation is 20 µm, similar to the average cell size the niche size has a large influence on the cell packing, dropping by more than 35% when the niche size goes from a sub-cellular size of 10 µm to a larger than the cell size of 30 µm, and the drop is even larger, for the larger niche sizes. If the wall separation is increased by just 5 µm to 25 µm, a little above the average cell size, the general cell packing drops. This suggests that the influence of the niche size decreases, as seen by the smaller spread of values for a 25 µm, as compared to the 20 µm wall separation. This tendency continues for larger wall separations, until the convergence point, where it seems the niche sizes have no more influence on the cell packing. The same observations are made for the cytoplasm material part in Figure 6b, where a slightly smaller drop in material per unit area is seen as a function of increasing scaffold dimension parameters.

This clear tendency of a less dense cell packing as the scaffold increases in both cases indicates that a larger influence on the cell arrangement by the scaffold is achieved when the cells are sensing the scaffold from all sides, i.e., when both niche size and wall separation are comparable to the average cell size. If one or both parameters gets larger than the average cell size the scaffold influence starts to diminish.

This is further explored in the following. We calculate for a given chamber size the ratio of the total cellular material compared to the amount of the cellular material found inside the niche, as shown in Figure 6c for the nuclei, and in Figure 6d for the cytoplasm contribution. In the analysis, the material found inside the niches, which is considered to be within a range up to 7.5 µm away from the walls, and further corresponds to roughly half the average cell size, is to be considered in contact with the walls. For each niche size there are 4 wall separations, which determine the chamber size, and therefore the ratio between the total area and the area covered by the niches. These ratios are 0.75, 0.6, 0.43 and 0.27 for wall separations of 20, 25, 35 and 55 µm respectively, and have been added to Figure 6c,d. In both figures, data points for an isotropic distribution have been added to guide the eye. For an isotropic distribution of cellular material in the chambers, and therefore a minimal influence of the scaffold’s presence on the cellular arrangement, the extract data points should match or be close to these curves.

For the nuclei contribution shown in Figure 6c, we observe that for a 10 µm niche size the ratio of nuclei material in contact with the nearly follows the predicted values for an isotropic distribution. This would indicate that the scaffold does not influence the nuclei packing significantly, which could be contributed to reaching a saturation of the number of cells able to be packed into the small volume of the 10 µm niche, even for larger wall separations. The three other niche sizes follow a general trend, which is different from the behavior observed for the 10 µm niche size. For the 20 µm and 25 µm wall separations the ratio of cells in the chambers compared to those within the niches, in proximity to the walls follow the predicted isotropic distribution values for the three niche sizes, while for larger wall distances the ratio is larger than the predicted values. This would indicate that the scaffold promotes a higher degree of nuclei packing in the niches even for large scaffold parameters giving rise to a non-isotropic cell distribution in the chambers. This could be promoted by the larger wall separation allowing for increased cell motility, and will be analyzed and discussed in the next section.

For a similar analysis on the cytoplasm part of the cells, seen in Figure 6d, the niche size of 10 µm, again follows the isotropic distribution behavior for all wall separations. For the three other niche sizes, a relatively constant ratio of cytoplasm localization in chambers versus within niches is observed as a function of the wall separation, which indicates that a high percentage of the cytoplasm part of the cells is in contact with the scaffold, even higher than for the nuclei part of the same cells.

### 3.4. Observation of 3D Cellular Vertical Growth over Time as a Function of Wall Separation

In the time-lapse studies, the scaffold is seeded with a lower total cell count than in the previous experiments, both to simplify tracking of the individual cells, as well as to quantify the effect of the scaffold from an initial situation, while limiting any scaffold saturation effects. We apply the same analysis tool, spectral linear unmixing, with a focus on the fluorescent contribution from the H33342 stained cell nuclei, to define the position of individual cells. Furthermore, the scaffold is separated into two parts in this analysis, a bottom part consisting of the first 14 µm in the z-direction, the average size of the A549 cell, and a top part with the remaining 10 µm of the scaffold height, as seen depictured in Figure 7a. From each of the 6 time points, a data set of the individual cell positions, identified by the cell nuclei, is recorded in both the top and bottom part of the scaffold. In the first part of the analysis, we only look at the influence of the wall separation. The influence of the niches is not taken into account, since the cell count for each niche size in the initial seeding stage, is too low for such statistical analysis.

Analysis of data for the first 14 µm of the scaffold close to the sample surface show minimal change in the cell count per unit area over time, as a function of the wall separation, even when compared to the flat 2D control surface, see Figure A2a in Appendix B. In the analysis, the unit area is chosen to be the area between two walls with a 20 µm separation. For the number of cells located inside the niches as compared to the total number of cells, no significant variations are observed again as a function of time, as depicted in Figure A2b in Appendix B.

The same analysis is performed on the 10 µm spanning top part of the scaffold, and similar non-changing behavior is found, with a few exceptions and modifications, see Figure A3a,b) in Appendix B. First, the number of cells in the control area, is by default low, since only non-attached floating cells in the growth media are counted in this region. Secondly, the overall number of cells at the top of the scaffold is lower, as compared to the bottom part, and cells are nearly all located inside the niches due to the lack of the 2D glass surface between the walls. Based on the extracted data from both the bottom and top of the scaffold, where the cell number does not change as a function of time, we do not consider proliferation as an influence in further analysis.

Analyzing the ratio of cells inside the niches as compared to the total number of cells, for both the top and bottom parts of the scaffold in the time-lapse study, is shown in Figure 7b. For the bottom 14 µm of the scaffold and wall separations of 20, 25 and 35 µm, the ratio of cells attached to the niches is between 0.74 and 0.97, as seen in the column in Table 1. While for a wall separation of 55 µm, only about half of the counted cells are attached to the scaffold, indicating that the wall separation has become large enough so that the A549 cells do not feel the immediate influence of the scaffold. For the 10 µm top part of the scaffold, wall separations of 20 and 25 µm, still have cell attachment ratios between 0.73 and 1, while for 35 µm separation the ratio has fallen to between 0.59 and 0.73, as seen in the second column in Table 1. For a separation of 55 µm, the ratio is again about half of the cells. This indicates that without the planar 2D surface between the walls, the cells stay floating in the medium, without attaching, if the cells do not come into direct contact with the wall. Comparing the values for the scaffold influence on the cell nuclei distribution in the low population scheme, Figure 7b, to the extracted nuclei part in Figure 6c for the full scaffold height in the strongly populated scaffold from Figure 5, as seen in the third column in Table 1, a general trend is observed.

For all wall separation distances the values for the low cell density case, have a higher ratio, indicating a stronger scaffold influence on the cells. As the cell number grows and the niches starts to fill, the influence of the scaffold lowers, reaching a saturation effect where the niche and chamber distribution is the same, as seen for the 10 µm niches in Figure 6c.

The constant number of cells between two walls for the 6 time points, for both regions of the scaffold, indicates that there is no net in- or out-flux of cells to the scaffold, and a limited vertical cell movement in the analyzed time period. The limited vertical movement could be due to the design of the scaffold, where the low height of the scaffold prevents a clear separation of the vertical movement between the two parts. This could be improved with a taller scaffold structure. Furthermore, the designed tight openings both to the outside of the scaffold and between chambers, as well as the wall structure itself, limit cell motility and only allow movement along the wall direction and not freely throughout the entire structure array. Analysis of this horizontal cell movement, as a function of time in the initial phase for both the top and bottom part of the scaffold, will be discussed further in the next section.

### 3.5. Analysis of Cell Movement Compared to Scaffold Wall Separation and Niche Size

Based on the results from time-lapse studies described in Section 3.4, where no changes in the total cell number over time and any vertical movement was observed, a more complex analysis of the individual horizontal cell motility is required to conclude the influence of the scaffold parameters. In our time-lapse study, each time point is spaced more than an hour apart, making tracking of the individual cells difficult. We, therefore, apply a custom written Matlab algorithm that, based on the nuclei position at each time point in the time lapse estimates the mobility of a cell, and from this, quantifies the horizontal movement in the two vertical sections in the scaffold area.

The algorithm starts by locating each cell, by the nuclei position in the first time point, t = 0 of the image series, and will, in each subsequent time point in the series, based on the single input parameter N, try to find the position of that cell in the next time point.

In each time point, a tracked nucleus is marked with a colored circle, orange in the first time point, etc., as seen in the tracking example for 6 time points in Figure 8a. In the current analysis when the algorithm has gone through the image sequence once, it again goes through the data set without already tracked cells and tries to map additional cells. If more than the two iterations of the algorithm used in the presented data is needed it is easily configured in the Matlab code. Furthermore, in our analysis we require that a tracked cell should be present in five sequential time points, which is based on the result from the previous section, where the cell count does not change over time. If the image sequence consists of more time points, or when a large number of cells are leaving the measured area, this number should be set accordingly in the Matlab code. After all possible cells have been tracked, the algorithm returns, based on the total movement of the tracked cells, a color gradient map of cell motility which is overlaid with the outline showing fold structure. A blue color, arrow 1 in Figure 8a, indicates stationary or low motility while large cell motility is indicated by a red color in the gradient map, see arrow 2 in Figure 8a). The lowered auto fluorescence of the scaffold, and the use of spectral linear unmixing makes the scaffold walls, and niches easily identifiable markers in Figure 8a for correlating cell motility to specific regions of the scaffold.

In the algorithm, the value of the parameter N defines a radius where a single cell nucleus has to be found in the next time point, thereby defining the maximum distance a cell can move between time points. N is given in pixels, and in our experimental setup with a 10× microscope objective, the conversion factor is a pixel = 1.66 µm. The algorithm selects a new cell position in the next time step, by choosing the cell closest to the current position if multiple cells are found inside a circle with a radius of N. If no cell is found inside N the cell will not be tracked. This also happens if the value of N is set to a low value compared to the motility of the cells monitored, whereby the cells have migrated beyond the radius of the circle defined by N. If the value N is set too high there is an increased risk that the algorithm will identify the wrong cell in the next time step, which also happens for a high concentration of cells in the scaffold area. The influence of the value of N for the current data set, can be seen in the extracted data in Table A1 in Appendix B and for the top and bottom part of the scaffold in Figure A4 and Figure A5, also in Appendix B. From data summarized in Table A1, one can observe that for N = 2 the highest total movement extracted is only 25% of the maximum allowed distance, while for N = 5 and N = 7 the movement increases to approx. 40% while for N = 10 and above it settles at approx. 30%. This is visible in Figure A4 and Figure A5, which represent the results of a mathematical analysis of cell center movements identified within confocal images collected at 6 different time points over an interval of max. 5 h 25 min. N is the number of pixels that define an allowed movement kernel from time point to time point. A gradient map is used to represent identified cell movement distances in dependence of the movement kernel size N, via a cell tracker Matlab code, see Appendix A. We superimpose the scaffold structure extracted by thresholding from the original confocal image on top of the gradient maps, as well as the various cell positions identified at the different time points (colored circles), as also done in Figure 8. ROIs (white dotted lines) were added that highlight areas of high cell movement, identified for the N = 10 gradient map and remaining similar even for higher movement kernel sizes (copied into the other gradient maps for reference).

We, therefore, apply N = 10 for further analyses, which is a value of 16.6 µm comparable to the average cell size, and slightly larger than the average cell motility of 10 µm/h found for A549 cells [29]. This value will optimize the number of cells analyzed since no two cells can occupy the same space in the scaffold, and give an individual cell motility range between 0 and 100 µm, in the time-lapse study. Figure 8b,c shows the gradient map for the bottom and top of the scaffold, respectively, for a value of N = 10 created by the algorithm, where both figures are scaled to a maximum total distance of 30 µm, the highest value found in the analysis.

The gradient map for the bottom part of the scaffold in Figure 8b, shows several regions of high motility i.e., orange to red regions [20–30 µm], which are characterized by large free spaces with flat 2D surfaces for unrestricted movement. The free spaces are either in the form of the planar 2D control region at the top of the figure or where the wall separation is 55 µm in combination with a niche size larger than the average cell size, i.e., 30, 50 and 70 µm. Other high motility regions are found where the niche sizes are large 50 and 70 µm, but also in regions where a large number of cells are located in a small region. At the other end of the motility spectrum, where the cells become very stationary seen as blue areas in Figure 8b, is where at least one scaffold parameter is equal to or below the average cell diameter.

A similar analysis is made for the top part of the scaffold, as seen in Figure 8c, where less horizontal cell motility is observed, indicating that the cells are more stationary when attached to the scaffold and there is no planar 2D surface to support them. Only when the wall separation becomes larger than the average cell size a few cells are found in the space between the walls.

The current analysis is limited to mapping cell movement in the XY-plane and does not extend to movement in the vertical Z-direction. For the scaffold in the present study with a small height, this limitation is not crucial while for extended scaffolds in Z-dimension, this movement would have to be taken into account.

## 4. Conclusions

We showed that a simple scaffold design combined with linear unmixing analysis has proven to be an excellent tool for cell-scaffold interaction studies. This approach can be used to separate the strong scaffold auto-fluorescence from the fluorescence signal from the labeled cells. To create such scaffolds, TPP have shown to be an ideal method of fabrication, where several size orders of magnitude can be spanned needed for the incorporation of 3D morphological hierarchy, thereby replicating the structural influence of the ECM on cells. While 3D TPP has been used to mimic ECM structures for example in the context of cell differentiation [30,31], and stem cell therapy [32,33], or to generate 3D microenvironments suited to conduct in vitro tests of medical therapies [34], here we propose the application of TPP to generate 3D metrology structures for the analysis of cell packing and cell motility. The systematically changed dimensions of wall separations and niches allow us to use the 3D polymer structures to “act as rulers“, that are able to operate on the typical dimensions of cellular sizes. The observed variation in cellular dynamics, allows us to identify niche sizes that promote high or low cell motility and may indicate preferred cellular growth conditions. The presented gradient scaffold design where structure parameters, such as wall separation and niche size, is easily varied throughout the scaffold, could prove to be a vital tool for fast analysis of optimal scaffold parameters for different cell types and co-cultures. Our analysis indicates that even if the scaffold has a simple design layout, the seeded cells show complex behavior which depends on several parameters. When scaffold parameters are below or similar to the A549 average cell size, cell packing is denser and the individual cells tend to be less mobile. On the other hand, it was also shown in our studies that these small features will quickly lose their influence on the incubated cells when the features are saturated. If one or more scaffold parameters, is larger than the average cell size, the A 549 cells tend to be more mobile and when attaching to the scaffold, do so in a less dense configuration not leading to saturation effects. While in the present study the scaffold design is deliberately kept as simple as possible a more complex 3D design would provide more detailed information about the cellular behavior on the scaffold, where especially the limited scaffold height in the current study could be extended. In this case, confocal imaging would not be ideal since its limited sample penetration capabilities, but could be done with standard or even advanced multiphoton imaging techniques with extended depth penetration [35]. Multiphoton imaging would also provide the possibility to do label free imaging of the seeded cells via metabolic markers [36]. For larger image data sets collected on 3D micro-ruler scaffolds interacting with cells deep learning approaches [37,38] may help extract further relevant information to help identify optimal cellular growth conditions for 3D cellular in vitro models.

## Figures and Tables

**Figure 1 polymers-14-04168-f001:**
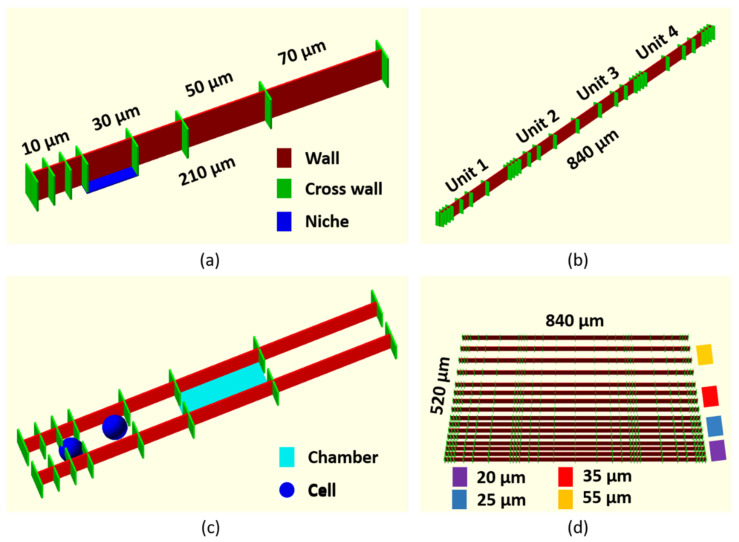
3D representation of the scaffold design. (**a**) A single 210 µm long and 24 µm tall unit wall (red), divided by 15 µm long cross walls (green) into different niche sizes, where an example is indicated (blue). (**b**) An 840 µm long wall structure created by 4 unit wall structures joint together. (**c**) Two unit wall structures, with a wall separation of 20 µm, creating chambers (light blue region) with openings marked by the cross walls. For size comparison two cells (blue spheres) with a diameter of 15 µm, are placed in the chambers. (**d**) The full scaffold where the 840 µm wall is replicated 16 times with 4 different wall separations, indicated by the different colors.

**Figure 2 polymers-14-04168-f002:**
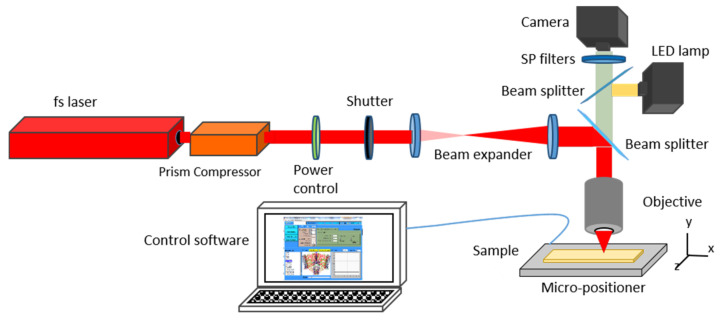
Schematic layout of the TPP fabrication setup.

**Figure 3 polymers-14-04168-f003:**
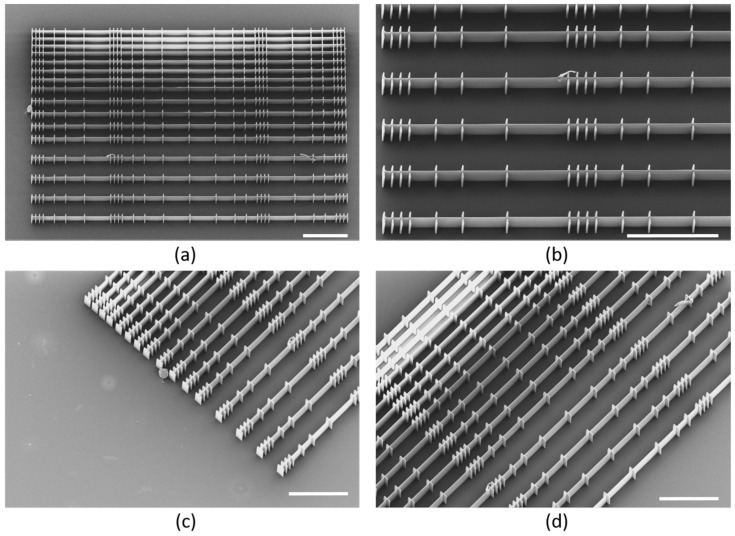
SEM images of scaffold structure, images (**a**,**b**) are at 20 degree tilting, while (**c**,**d**) are at 25 degrees. (**a**) Full scaffold. (**b**) Zoom of the unit structures, showing the 10, 30, 50, and 70 µm niche sizes, here with a 55 µm wall separation. (**c**) Zoom of scaffold end facet, showing the different opening size between the cross walls. (**d**) Rotated and angled view showing the different wall separation of 20, 25, 35, and 55 µm. Scale bars 100 µm.

**Figure 4 polymers-14-04168-f004:**
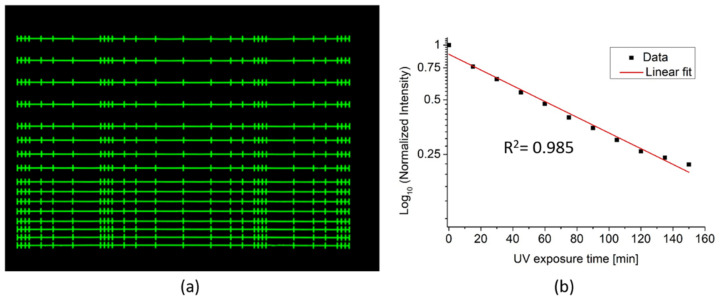
Quenching of scaffold auto-fluorescence via UV light treatment. (**a**) Confocal scan of the scaffold at starting conditions. (**b**) Analysis show the quenching follows an exponential behavior as a function of time, and the 150 min treatment results in a nearly 80% reduction of the auto-fluorescence.

**Figure 5 polymers-14-04168-f005:**
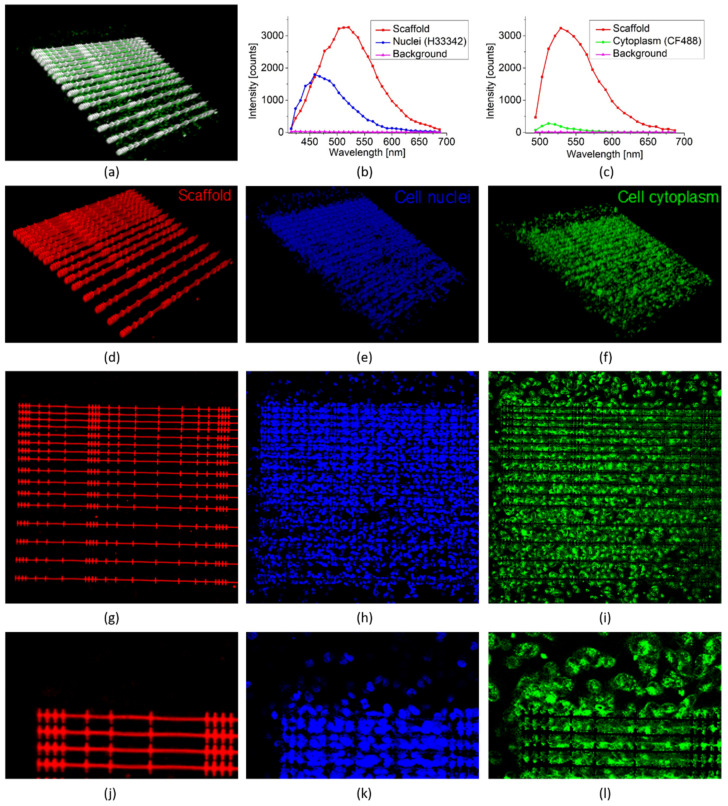
3D hyperspectral confocal fluorescence image of the scaffold incubated with A549 cells and results of spectral un-mixing. (**a**) 3D hyperspectral image—polymer with broad emission spectrum and partly saturated appears white, cells in green (exc. 405/all emission channels integrated). (**b**) Spectra associated to selected image regions of the scaffolds, cell nuclei and background areas, with excitation @405 nm and 32 detection channels. (**c**) Spectra associated to selected image regions of the scaffolds, cell cytoplasm and background areas, with excitation @488 nm and 23 detection channels. (**d**–**f**) Linear unmixing result of 3D areas identified to match the spectrum associated to the scaffold, cell nuclei and cytoplasm, respectively. (**g**–**i**) Representative image slice showing the quality of the linear unmixing. (**j**–**l**) Zoom images, showing details of the linear un-mixing.

**Figure 6 polymers-14-04168-f006:**
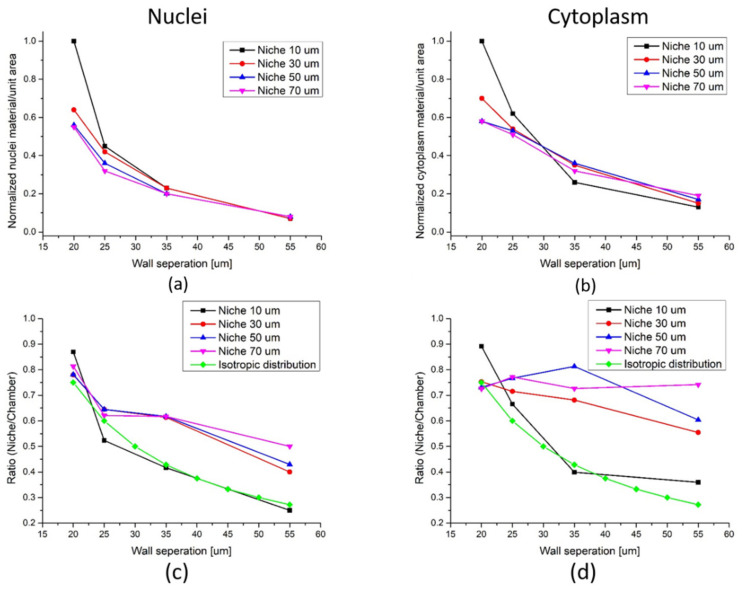
Statistical analysis of cell localizations within the scaffold. Total amount of fluorescent cellular material per area, as a function of niche size and wall separation, (**a**) nuclei and (**b**) cytoplasm. Ratio of total cellular material found in a given chamber to cellular material found inside the niches, i.e., in contact with the walls in the given chamber, (**c**) nuclei (**d**) cytoplasm.

**Figure 7 polymers-14-04168-f007:**
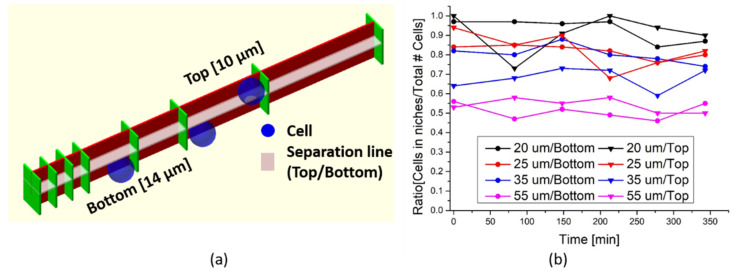
(**a**) Schematic representation of the definition of the bottom of the scaffold, below the transparent line and the top, above the transparent line. The 3 blue spheres are examples of the average cell size, where 2 will be counted in the bottom and one would be counted as belonging to the top. (**b**) Analysis of the ratio of cells attached to the scaffold as compared to the total number of cells.

**Figure 8 polymers-14-04168-f008:**
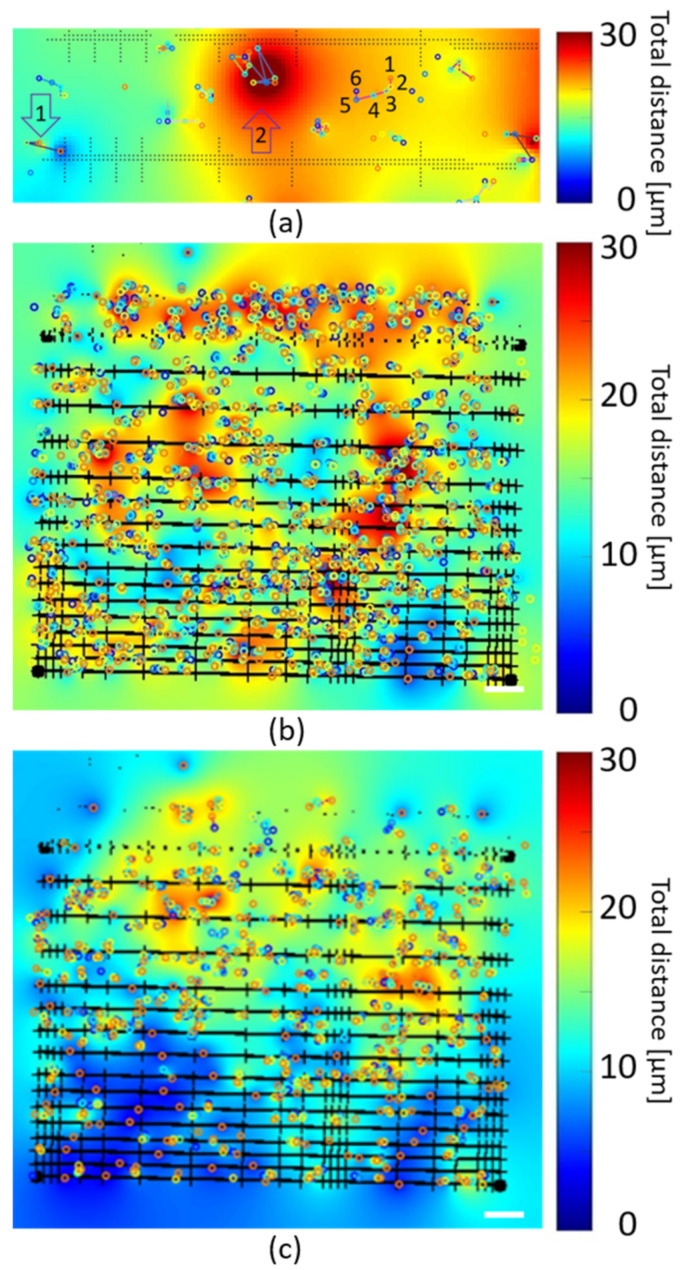
Gradient maps showing cell movement of time-lapse study with movement kernel parameter N = 10 (**a**) Zoomed image showing detailed movement of a single cell in the 6 time points (1–6), an area with low cell movement (arrow 1) and high motility (arrow 2), as well as the scaffold structure. (**b**) Bottom part of the scaffold. (**c**) Top part of the scaffold.

**Table 1 polymers-14-04168-t001:** Ratio of cells inside the niches as compared to the total number of cells, for initial seeding phase (Figure 7b and a densely populated scaffold Figure 6c).

Wall Separation	Bottom (Figure 7b)	Top (Figure 7b)	Figure 6c
20 µm	[0.84–0.97]	[0.73–1.00]	[0.78–0.86]
25 µm	[0.76–0.85]	[0.76–0.94]	[0.52–0.64]
35 µm	[0.74–0.88]	[0.59–0.73]	[0.41–0.62]
55 µm	[0.46–0.56]	[0.50–0.58]	[0.25–0.50]

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
