# Peer review of "3D Polymer Architectures for the Identification of Optimal Dimensions for Cellular Growth of 3D Cellular Models"

_polymers, 2022, doi:10.3390/polym14194168_

Round 1
Reviewer 1 Report
The paper from Maibohm et al. investigates the cellular growth in custom-made scaffolds by using two-photon polymerization technique. The flexibility of the technique seems to allow a fast and high-throughput study of the influence of structural parameters on cell distribution.
In my opinion, this work is of wide interest because it is not yet well known the effect of geometry on cell proliferation and a simple geometry, and a consistent technology can help to understand more complex scaffolds geometries. In my opinion, the work could be published after a revision. I would like the authors to consider the following comments before recommending publication in Polymers journal:
1. How do the authors extract the features from the images of cell proliferation? I think that since there are several variables, a simple manual approach with ImageJ is leaving information out. Why the authors do not consider Deep learning?
2. I would suggest introducing the error (e.g. standard deviation or instrument error) on the measurement done on niche sizes, cell size and wall.
3. Another techniques that allows to produce morphological hierarchy in 3D is 3D foam printing (i.e. https://doi.org/10.1002/adem.202101226). It is cost and time efficient compared to polymerization techniques but maybe with less resolution. Have the authors considered other 3D technique for their study? I would mention them in the introduction.
Minor-corrections and typos:
Typos at lines 575 and 576 “Title” missing.
Reviewer 2 Report
Authors presented a work focused on the fabrication of 3D-scaffolds based on bio-compatible polymer SZ2080 via two-photon polymerization and reported the factors/parameters affecting their effects on A549 lung epithelia cells growth and movement.
I would suggest to include more recent works (2021-2022) on the bibliographic work
Concerning figure 5 fluorescence. Authors tried to superposed image with red/blue/green (scaffold/nuclei/cytoplasm)? Often such superposition bring interesting information.
I would suggest o include a table/paragraph highlighting the previous work, and contrasting them with their main findings
Authors presented this work was partially presented in the form of a proceeding; this should be included. https://doi.org/10.3390/Micromachines2021-09596. Authors’ must ensure that material presented in the present works is fully unpublished
Figures A3 and A4 are hard to follow. Authors can include arrows? Or explanation which will be very helpful for readers.
Round 2
Reviewer 1 Report
The authors improved their manuscript and I have no further comments.
Reviewer 2 Report
authors responded reviewer comments